# Mutually exclusive sense–antisense transcription at *FLC* facilitates environmentally induced gene repression

Stefanie Rosa[1], Susan Duncan[1] & Caroline Dean[1]

Antisense transcription through genic regions is pervasive in most genomes; however, its functional significance is still unclear. We are studying the role of antisense transcripts (*COOLAIR*) in the cold-induced, epigenetic silencing of Arabidopsis *FLOWERING LOCUS C* (*FLC*), a regulator of the transition to reproduction. Here we use single-molecule RNA FISH to address the mechanistic relationship of *FLC* and *COOLAIR* transcription at the cellular level. We demonstrate that while sense and antisense transcripts can co-occur in the same cell they are mutually exclusive at individual loci. Cold strongly upregulates *COOLAIR* transcription in an increased number of cells and through the mutually exclusive relationship facilitates shutdown of sense *FLC* transcription in *cis*. *COOLAIR* transcripts form dense clouds at each locus, acting to influence *FLC* transcription through changed H3K36me3 dynamics. These results may have general implications for other loci showing both sense and antisense transcription.

[1] Department of Cell and Developmental Biology, John Innes Centre, Norwich Research Park, Norwich NR4 7UH, UK. Correspondence and requests for materials should be addressed to C.D. (email: caroline.dean@jic.ac.uk).

There is considerable interest in understanding the mechanisms through which antisense transcription functions in gene regulation[1,2]. A classic example of environmentally mediated regulation involving antisense RNAs is vernalization, which in Arabidopsis involves the cold-induced epigenetic silencing of *FLOWERING LOCUS C* (*FLC*), a repressor of flowering[3,4]. An early phase in vernalization is the upregulation of a set of antisense transcripts, named *COOLAIR* that encompass the entire *FLC* locus[5]. Before cold *COOLAIR* transcription is repressed by the NDX1 homeodomain protein, which stabilizes an R-loop (where the nascent transcript invades the DNA helix) over the *COOLAIR* promoter[6]. The cold-induced *COOLAIR* transcription accelerates repression of *FLC* transcription[7], but the mechanism through which this occurs is not known. As cold exposure continues, epigenetic silencing of *FLC* quantitatively increases, progressively reducing *FLC* re-activation when the plants return to warm conditions. This quantitative memory is due to a switch of epigenetic state at *FLC* in an increasing fraction of cells[8]. This switch is cell-autonomous, mediated by a modified Polycomb repressive complex 2 with the local chromatin environment of each locus instructing its own epigenetic inheritance[9]. This mechanism involves *COOLAIR*-dependent switching of opposing histone modification states (H3K36me3 to H3K27me3) and spreading of Polycomb silencing across the locus to maintain long-term memory of the silenced state[5,8,10,11].

To elucidate the role of *COOLAIR* in this Polycomb switching mechanism we used single-molecule fluorescence *in situ* hybridization (smFISH) to detect individual sense and antisense RNAs. We show that *COOLAIR* transcription is anti-correlated with *FLC* transcription and unspliced *COOLAIR* forms dense clouds around the *FLC* locus in the early weeks of cold. The nature and timing of these events reveals the importance of *COOLAIR* to promote efficient *FLC* transcriptional repression before Polycomb silencing.

## Results

**smFISH for cellular analysis**. We adapted a smFISH method[12,13] to visualize and quantify single RNA transcripts in plant cells. The developmental structure and transparency of the Arabidopsis root made it an ideal tissue for this procedure. *FLC* messenger RNA (mRNA) transcripts showed a graded reduction in each cell as the duration of cold exposure increased (Fig. 1a–c). This was not a general cold-induced reduction, as levels of a constitutively expressed gene *Protein Phosphatase 2A* (*PP2A*) remained high after 6-weeks cold exposure (Supplementary Fig. 1). Expression determined from this approach corresponded well to quantitative reverse transcription–PCR measurements of *FLC* mRNA (Supplementary Fig. 2a).

In contrast to the graded and slow reduction in number of *FLC* mRNA molecules, *FLC* non-spliced transcripts (representing transcription) decreased quickly with many cells showing no signal after 2-weeks of cold (Fig. 1a,b,d). No obvious mRNA re-localization was revealed, as the mRNA signals showed no change in distribution before or after cold (Fig. 1b; Supplementary Fig. 2b). These observations suggest that dynamic molecular changes at the locus rapidly turn off *FLC* transcription in response to cold.

**COOLAIR accumulates over FLC loci after cold exposure**. Since *COOLAIR* is activated early during cold, we asked whether it has a role in this process. Intronic probes that detected non-spliced transcript (Fig. 2a) revealed signal in only ~3% of total cells in warm-grown plants (Fig. 2b), explaining the overall low levels of *COOLAIR* expression[5]. However, after 2-weeks of cold exposure,

non-spliced *COOLAIR* increased in many cell types in the root, such that overall the number of cells expressing *COOLAIR* increased to ~30% (Fig. 2b). In addition, *COOLAIR* foci increased in intensity and number in each nucleus (Fig. 2c,e; Supplementary Fig. 3). The distribution was reminiscent of nuclear bodies implicated in processing and maturation of ribonucleoprotein complexes and aberrant mRNAs[14,15]. Sequential smFISH-immunolabelling of both non-spliced *COOLAIR* and the Cajal body marker U2B' in the same cell, however, showed no co-localization (Supplementary Fig. 4). To understand the origin of the different foci we performed a sequential RNA–DNA FISH in cold-exposed root cells. This showed the large but not the small *COOLAIR* foci co-localized with *FLC* DNA FISH signals (Fig. 2d, Supplementary Fig. 5). The larger accumulations of *COOLAIR* are thus in 'clouds' covering each *FLC* locus. We have previously detected *COOLAIR*-chromatin interactions using chromatin isolation by RNA purification[7]. However, these correspond to localized and tightly integrated structures such as the R-loop at the *COOLAIR* promoter[6]. The *COOLAIR* 'clouds' detected by FISH appear to be a more general association of the *COOLAIR* non-spliced transcript with *FLC* chromatin. Both types of data support a cis-regulatory function for *COOLAIR* transcription/transcripts in *FLC* regulation. Double labelling with *COOLAIR* exonic and intronic probes revealed that the large *COOLAIR* clouds invariably labelled with both probe sets and thus represent non-spliced transcript at the locus (Supplementary Fig. 6a,c). In the large majority of cells the small foci showed intronic signals only, however, occasionally we also observed small foci labelling with exonic probes alone (Supplementary Fig. 6c). We speculate that these smaller foci might correspond mainly to intron 1 lariats dispersed throughout the nucleus before degradation. Probes to *COOLAIR* exonic sequences did not reveal signals in the cytoplasm (Supplementary Fig. 6b). We have previously found spliced *COOLAIR* enriched in cytoplasmic fractions[7], so the absence of cytoplasmic signal in the FISH may be due to the extensive secondary structure of the spliced *COOLAIR* transcripts[16]. *COOLAIR* levels were higher in *ndx1-1,* a mutant defective in the homeodomain protein that represses *COOLAIR* transcription through stabilization of the R-loop at the *COOLAIR* promoter[6] supporting the view that cold and NDX1 regulate *COOLAIR* transcription independently (Supplementary Fig. 7).

To understand the basis of the accumulation of the *COOLAIR* clouds at the *FLC* locus, we generated probes specific for either the 5′ or 3′ end of the *COOLAIR* largest intron (Fig. 2f). In some cells the 5′ set gave a higher signal, whereas in others the 5′ and 3′ sets gave similar signals (Fig. 2g,h, Supplementary Fig. 8), which we interpret to indicate the clouds are likely to be formed from both increased transcription and transcript accumulation at the locus. More detailed analysis of *COOLAIR* transcription rate and turnover will be required to fully elucidate this. The co-localization of *COOLAIR* 5′ and 3′ probe set provides an important control for off-target signals validating the specificity of the probe sets used, similar to the two-colour approach used by Cabili *et al.*[17]. A similar control with *FLC* 5′ and 3′ intronic probes (Supplementary Fig. 2c) and co-localization in the DNA–RNA FISH confirmed the specificity of the smFISH probes (Supplementary Fig. 2d).

**COOLAIR and FLC transcription are mutually exclusive**. We then asked whether formation of *COOLAIR* clouds would affect *FLC* transcription at the same locus. We undertook a thorough analysis of the accumulation of *COOLAIR* and *FLC* in single cells over a time course of cold exposure (Fig. 3a). Although *FLC* sense

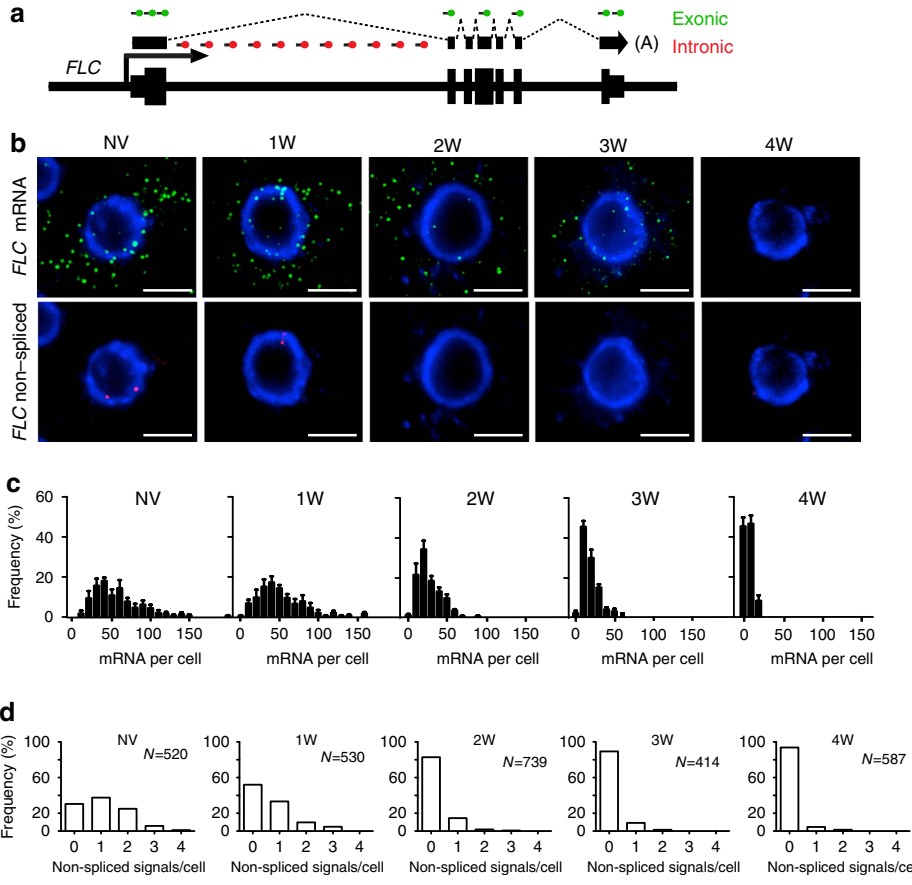

**Figure 1 | Single-cell quantification of mRNA expression and *FLC* transcription during vernalization.** (**a**) Schematic of the probes used to detect *FLC* transcripts: intronic (red) and exonic (green). (**b,c**) Cytoplasmic *FLC* mRNA decreases gradually in the cold. (**b**) Representative images of cells showing the gradual decrease of *FLC* mRNA (green) and the reduction in nuclear non-spliced *FLC* transcripts (red) in the cold. DNA labelled with DAPI (blue). Scale bar, 5 μm. (**c**) Frequency distribution of *FLC* mRNA molecules per cell in non-vernalized plants (NV) and after different weeks of cold (1W, 2W, 3W, 4W). (**d**) Quantification of active sites of transcription as judged by non-spliced signal at the time points depicted in **c**. *N* = number of cells analysed. All error bars are +/− s.e.m.

and antisense transcripts could co-occur in single cells (Fig. 3c), they were always transcribed from different alleles at any given time. Sense and antisense expression at individual loci were anti-correlated in both non-vernalized and cold-exposed cells (Fig. 3b), indicating transcription at the locus is unidirectional at any given time. This supports the view that *COOLAIR* is retained at the locus from which it is transcribed and acts in cis to regulate *FLC*. Analysis of *FLC* and *COOLAIR* expression in different genotypes had previously shown a positive correlation in expression[5,18]. The apparent contradiction between smFISH and the whole cell population data is likely due to mutually exclusive transcription of the sense or antisense strand both similarly influenced by trans factors and the local chromatin environment. Cold exposure would induce additional *COOLAIR* transcription at most loci, facilitating shutdown of *FLC* transcription and breaking the correlation.

To test this hypothesis, we asked whether this sense–antisense mutual exclusivity had implications for *FLC* transcription. We exploited the expression of *COOLAIR* enriched at vasculature precursor cells (Fig. 4a; Supplementary Fig. 9) and asked if the extent of *FLC* repression depended on the presence of *COOLAIR*. After 2-weeks of cold *FLC* expression had decreased ∼20% in non-*COOLAIR* expressing cells (Fig. 4c) and ∼75% in *COOLAIR* expressing cells (Fig. 4d). To confirm this difference was due to *COOLAIR* accumulation we analysed *FLC* transcriptional downregulation in a line (*FLC-TEX*)[7] defective in cold

induction of *COOLAIR* (Fig. 4b). *FLC* sense transcription was less effectively downregulated in this line after 2-weeks of cold (Fig. 4d). Importantly, this effect was only observed in *COOLAIR* expressing cells, as no change in *FLC* repression was observed in non-*COOLAIR* expressing cells (Fig. 4c). These data indicate that the accumulation of *COOLAIR* accelerates *FLC* repression during cold exposure.

## Discussion

Our ability to analyse *COOLAIR* at the single cell level has revealed that an early step in vernalization is the accumulation of *COOLAIR* transcripts at the *FLC* locus. *COOLAIR* upregulation coincides with the disruption of a gene loop at *FLC*[19] and accelerates sense transcriptional shutdown within 1–2 weeks as compared with the 4 weeks required for maximal intragenic Polycomb nucleation[8]—(Fig. 4e). The anti-correlation of *COOLAIR* and *FLC* transcription at the gene level would seem to exclude a cis-based colliding polymerase transcriptional interference mechanism. There is also no evidence for double-stranded RNA and the RNA interference machinery being required for vernalization[20]. We, therefore, propose that both the process of antisense transcription, and the presence of clouds of cis-tethered transcript, effectively repress *FLC* transcription. Indeed, disruption of *COOLAIR* predominantly affects the cold-induced reduction of H3K36me3, the opposing chromatin mark to the

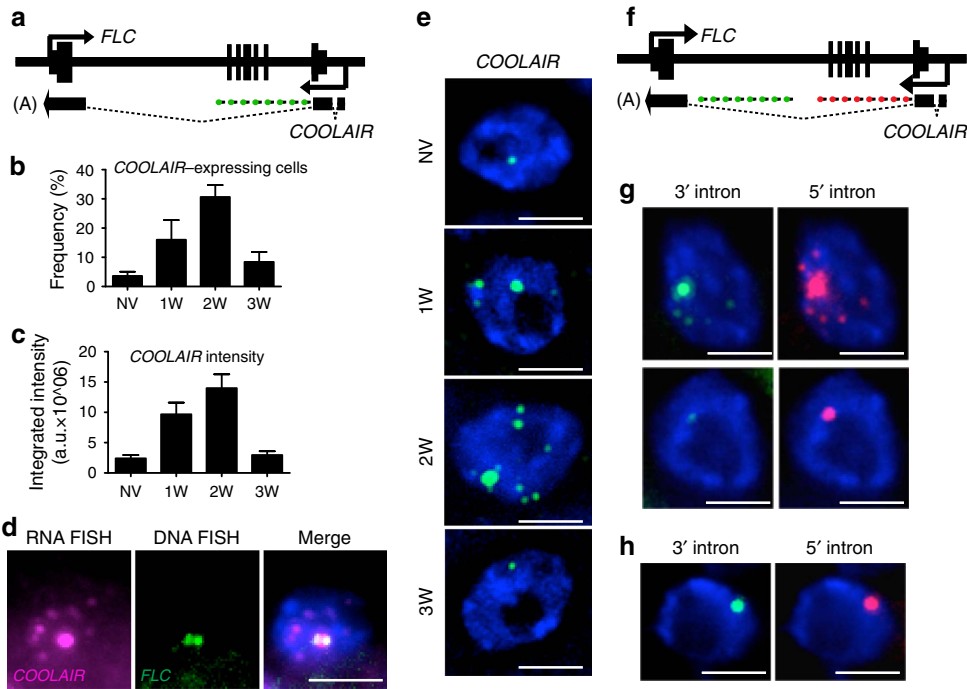

**Figure 2 | COOLAIR transcription rapidly increases in some cells in the cold and coats the locus.** (**a**) Schematic of the probes used to detect COOLAIR non-spliced transcripts. (**b**) Frequency of cells showing COOLAIR signals with increasing cold exposure (non-vernalized (NV), 1W, 2W, 3W). Error bars are $+/-$ s.e.m., with $N = 250$ cells. (**c**) Quantification of total integrated intensity of the large COOLAIR foci at different time points with increasing cold exposure. Error bars are $+/-$ s.e.m., with $N = 60$ cells. (**d**) Representative image of a nucleus showing co-localization of COOLAIR non-spliced RNA smFISH signals (magenta) with FLC DNA FISH signals (green). (**e**) Representative images of nuclei hybridized with COOLAIR smFISH probes (green), depicted in **a**, in NV and plants exposed to different weeks of cold (1W, 2W, 3W). The data show the increase in the number and intensity of COOLAIR foci per nucleus. (**f**) Schematic of the probes used to detect the 5′ (red) and 3′ (green) end of COOLAIR largest intron. (**g**) Representative images of nuclei showing higher signals at the 5′-end, in plants exposed to 2-weeks of cold. (**h**) Representative image of a nucleus showing equally high signals at the 5′ and 3′ end, in plants exposed to 2-weeks of cold. DNA labelled with DAPI (blue). Scale bar, 5 μm.

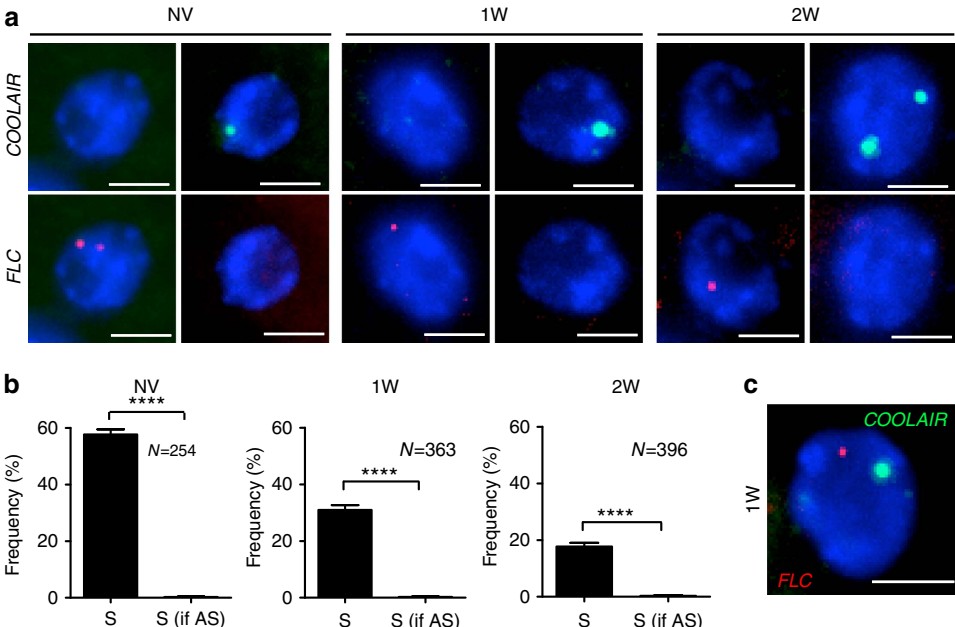

**Figure 3 | FLC and COOLAIR transcription are anti-correlated at the level of the locus.** (**a**) Representative images of nuclei hybridized with intronic smFISH probes for COOLAIR (green) and FLC (red) showing mutually exclusive transcription. Plants were given no cold (non-vernalized plants (NV)) or different weeks of cold (1W, 2W). DNA labelled with DAPI (blue). Scale bar, 5 μm. (**b**) Frequency of loci transcribing FLC, when COOLAIR is not present (S) or is detectable at the same locus (S if AS). FLC and COOLAIR transcription are anti-correlated at the level of the gene throughout all time points analysed (NV, 1W, 2W). $N =$ number of cells analysed. All error bars are $+/-$ s.e.m. (**c**) Representative image of a nucleus hybridized with intronic smFISH probes against non-spliced COOLAIR (green) and FLC (red) in plants exposed to 1-week cold (1W), showing transcription of FLC and COOLAIR from different alleles in the same cell. Scale bar, 5 μm.

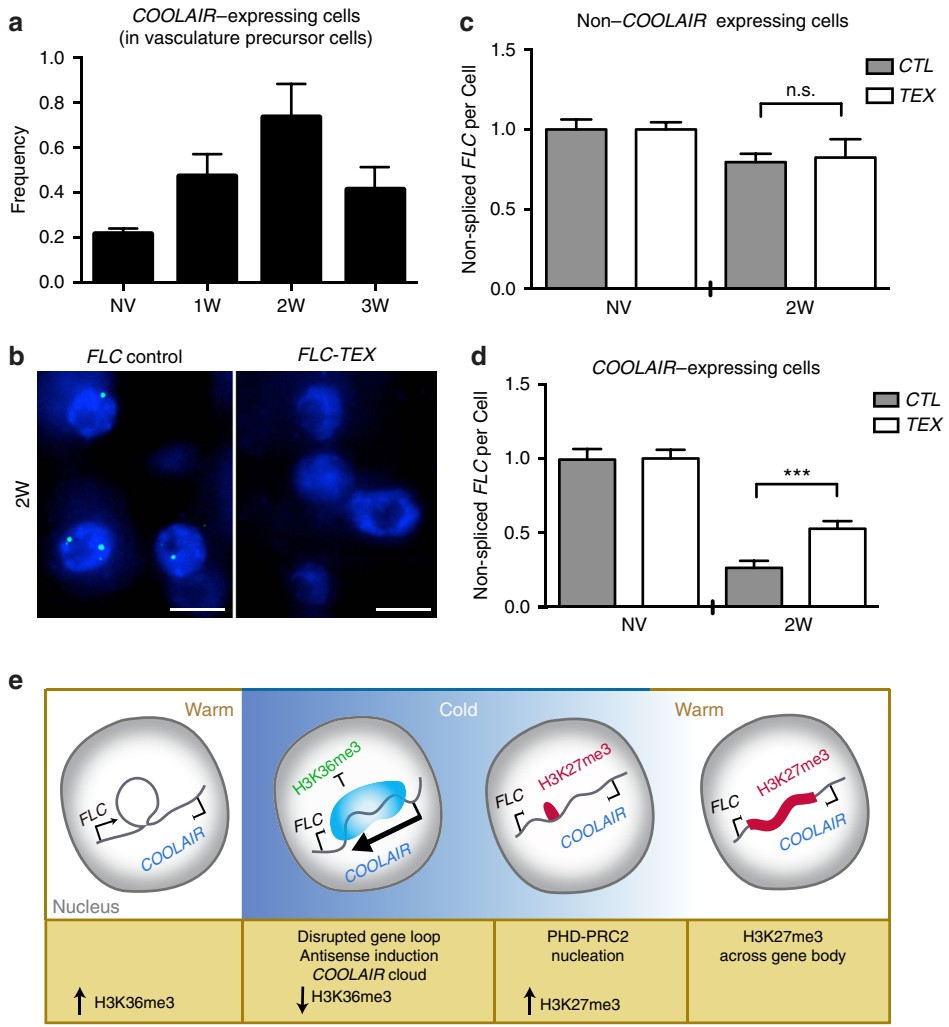

**Figure 4 | *COOLAIR* facilitates shutdown of *FLC* sense expression.** (**a**) Frequency of vasculature precursor cells showing *COOLAIR* transcription after increasing cold exposure (non-vernalized (NV), 1W, 2W, 3W). Error bars are $+/-$ s.e.m., with $N = 200$ cells. (**b**) Representative images of nuclei hybridized with intronic smFISH probes against *COOLAIR* (green) in plants vernalized for 2 weeks (2W), in control transgenic plants carrying an intact *FLC* gene and plants carrying an *FLC* terminator exchange construct lacking *COOLAIR* promoter (*FLC-TEX*). Scale bar, 5 μm. (**c**) Quantification of non-spliced *FLC* transcripts per cell, in non-*COOLAIR* expressing cells from control transgenic plants (*CTL*) and a *FLC-TEX* line (*TEX*), in NV plants and plants vernalized for 2 weeks (2W); frequencies were normalized to NV values. Error bars are $+/-$ s.e.m., with $N = 320$ cells. (Student's *t*-test, n.s., not-significant) (**d**) Quantification of non-spliced *FLC* transcripts per cell, in *COOLAIR* expressing cells from control transgenic plants (*CTL*) and a *FLC-TEX* line (*TEX*), in NV plants and plants vernalized for 2 weeks (2W); frequencies were normalized to NV values. Error bars are $+/-$ s.e.m., with $N = 300$ cells (Student's *t*-test, ***$P < 0.001$). (**e**) Schematic representing the current understanding of the role of *COOLAIR* in vernalization.

Polycomb-deposited H3K27me3, at the nucleation region in *FLC*[7,11]. *COOLAIR* transcription/transcripts could trigger a facultative heterochromatin state, potentially through blocking the activity/recruitment of H3K36me3 modification complexes, or enhancing recruitment of H3K36me3 demethylases, to maintain repression of *FLC* transcription as a necessary step to enable Polycomb-induced epigenetic switching[21,22].

Our analysis reveals that *COOLAIR* activity differs in many ways from well-characterized antisense transcripts in *S. cerevisiae*, such as from *PHO84* antisense that is continuously transcribed at a low level with polyadenylated antisense transcripts rapidly exported to the cytoplasm[23], or from the *GAL10* locus where sense and antisense transcripts are transcribed simultaneously from the same locus[24]. Conversely, *COOLAIR* shows many parallels with the non-coding Xist RNA involved in mammalian X-inactivation[25]; timed expression, cis-localization to a silenced allele and activity as a prelude to Polycomb silencing[26]. These parallels suggest that antagonistic sense-antisense transcription

might be a core mechanism underlying the evolution of many epigenetic mechanisms.

## Methods

**Plant material and growth conditions.** Seeds were stratified for 3 days at 5 °C before germination in a growth cabinet (Sanyo MLR-351H) in vertically oriented petri dishes containing Murashige and Skoog (MS) media minus glucose (16 h light, 100 μmol m$^{-2}$ s$^{-1}$, 22 °C ± 1 °C). Seeds were surface sterilized in 5% v/v sodium hypochlorite for 5 min and rinsed three times in sterile distilled water.

The lines used in this study were described, previously: ColFRI (ref. 27), *FLC-TEX*[7], *FLC* transgenic[7], *ndx1-1FRI* (ref. 6).

**Synthesis of the probes.** The probes were chosen with the following criteria: 1) they should be roughly 17–22 bases long, 2) their GC content should be as close to 45% as possible and 3) there should at least three nucleotides of space between their target regions. We used the online program Stellaris Probe Designer version 2.0 from Biosearch Technologies. For probe sequences see Supplementary Tables 1–8.

**smFISH procedure on root squashes.** Seedlings were removed from the media and the root tips were cut using a razor blade and placed into glass wells containing 4% paraformaldehyde (PFA) and fixed for 30 min. Roots were then removed from fixative and washed three times with nuclease free 1× PBS (Thermo Scientific, Lutterworth, UK). Several roots were then arranged on poly-L-lysine slide (Thermo Scientific, Lutterworth, UK) and covered by a glass cover slip (Slaughter, Uppminster, UK). The meristems were then squashed by tapping the cover slips, and immediately after each slide was submerged (together with the cover slip) for a few seconds in liquid nitrogen until frozen. The cover slips were then flipped off the slides using a razor blade and the roots were left to dry at room temperature for 30 min.

Tissue permeabilization was achieved by immersing the samples in 70% ethanol for a minimum of 1 h. The ethanol was then left to evaporate at room temperature for 5 min before two washes were carried out with wash buffer (containing 10% formamide and 2× SSC). One-hundred microlitres of hybridization solution (containing 10% dextran sulfate, 2× SSC and 10% formamide) with each probe set at a final concentration of 250 nM was then added to each slide. Cover slips (Slaughter, Uppminster, UK) were carefully laid over the samples to prevent the evaporation of the buffer and the probes were left to hybridize at 37 °C overnight in the dark.

The hybridization solution containing unbound probes was pipetted off in the morning. Each sample was washed twice with wash buffer with the second wash left to incubate for 30 min at 37 °C. One-hundred microlitres of the nuclear stain 4,6-diamidino-2-phenylindole (DAPI; 100 ng ml⁻¹) was then added to each slide and left to incubate at 37 °C for 30 min. The DAPI was removed and 100 µl 2× SSC was added and then removed. One-hundred microlitres GLOX buffer minus enzymes (0.4% glucose in 10 mM Tris, 2× SSC) was added to the samples and left to equilibrate for 2 min. This was removed and replaced with 100 µl of GLOX buffer containing enzymes (glucose oxidase and catalase). Glucose oxidase (#G0543 from Sigma) and catalase (#C3155 from Sigma), 1 µl of each enzyme is added to a total of 100 µl of GLOX minus enzymes. The samples were then covered by 22 × 22 mm No.1 cover slips (Slaughter, Uppminster, UK), sealed with nail varnish and immediately imaged.

**Combined smFISH with immunofluorescence.** We performed a sequential smFISH-immunofluorescence by first performing the RNA FISH using smFISH probes using the protocol outlined above and then performing the immunofluorescence. For sequential immunofluorescence-smFISH, cover slips were unmounted after imaging RNA FISH and washed three times in 4× -SSC/0.2% Tween at 37 °C. After slides were re-fixed with 4% (w/v) for-maldehyde freshly made from PFA in PBS buffer for 10 min. After washing in PBS for 5 min, the slides were digested in a mixture of 1% dolicalase, 0.5% cellulase, 0.025% pectolyase) at room temperature for 10 min. After enzyme treatment, roots were washed in PBS three times for 5 min each. Then the slides were blocked with 4% BSA in PBS for 50 min, followed by incubation with primary antibody. Anti-U2B' antibody from mouse (1:20) (gift from Peter Shaw[28]) was used as a primary antibody in 1% BSA in PBS. The slides were incubated at 37 °C for 1 h in a humid box. After washing the slides in PBS for 5 min each, anti-mouse IgG A488 conjugate (Invitrogen) was applied as secondary antibody (1:200) and slides were incubated for 1 h in a humid dark box at 37 °C. After washing the slides three times in PBS for 5 min each, the specimens were counter- stained with 1 g ml⁻¹ DAPI (49,6-diamidino-2-phenylindole dihydrochloride hydrate; Sigma) and mounted in Vectashield (Vector Lab). In order to find the cells previously imaged for smFISH, we saved the stage positions of those cells at the microscope and acquired large image tiles in order facilitate the identification of the cells.

**Combined RNA–DNA FISH.** For sequential RNA–DNA FISH, cover slips were unmounted after imaging RNA by smFISH (using the protocol outlined above) and washed three times in 4× -SSC/0.2% Tween at 37 °C. After that slides were re-fixed with 4% (w/v) formaldehyde freshly made from PFA in 1× PBS buffer for 10 min and washed again several times in 1× PBS. Slides were treated with 100 µg ml⁻¹ RNase for 1 h at 37 °C and washed twice in 1× PBS. After washing slides were digested in a mixture of 1% driselase, 0.5% cellulase, and 0.025% pectolyase for 10 min at 37 °C. Slides were then washed and re-fixed with 4% (w/v) formaldehyde freshly made from PFA in PBS buffer for 10 min. Slides were then transferred to a series of ethanol steps increasing to 70, 85 and 100%.

Probes were labelled with Digoxigenin-11-dUTP (#11745816910, Roche) by nick translation. Bacterial artificial chromosome clone JAtY71K18, which contains an insert of 75 kb, was used as a probe. The hybridization mixture (20 ng ml⁻¹ labelled DNA, 50% formamide, 10% dextran sulfate, 2× SSC, 1 mg ml⁻¹ salmon sperm (D9156, Sigma)) was denatured at 85 °C for 10 min and applied to the slides. Slides containing the hybridization mixture were denatured for 7 min at 75 °C (in a omnislide), and hybridized overnight at 37 °C. After hybridization, slides were washed at 42 °C once in 2× SSC, twice in 20% formamide plus 0.1 × SSC and twice in 2× SSC, and then twice in 2× SSC at room temperature and twice in 4× SSC plus 0.2% Tween-20. Then the slides were blocked in TNB (0.1 M TrisHCl, 0.15 M NaCl, 3% BSA) for 30 min at 37 °C. Digoxigenin-2-deoxyuridine, 5-triphosphate probes were detected with anti-DIG-fluorescein antibody (#11207741910, Roche) prepared in TNB buffer (1:100). Nuclei were counterstained with 1 mg ml⁻¹ DAPI, and slides were mounted in Vectashield (Vector Laboratories). In order to find the cells previously imaged for smFISH, we saved the stage positions of those cells at the microscope and acquired large image tiles in order facilitate the identification of the cells.

**Image acquisition.** A Zeiss Elyra PS1 inverted microscope was used for imaging. A ×100 oil-immersion objective (1.46 NA) and cooled electron multiplying-CCD (charge-coupled device) Andor iXon 897 camera (512 × 512 QE > 90%) was used to obtain all images in the standard, rather than super-resolution mode. The following wavelengths were used for fluorescence detection: for probes labelled with Quasar570 an excitation line of 561 nm was used and signal was detected at 570–640 nm; for probes labelled with Quasar670 an excitation line of 642 nm and signal was detected at 655–710 nm; for DAPI an excitation line of 405 nm and signal was detected at wavelengths of 420–480 nm. For all experiments series of optical sections with z-steps of 0.2 µm were collected.

Z-stacks were deconvolved using AutoQuant X2 (Media Cybernetics). Maximum projections and analysis of three-dimensional pictures were performed using Fiji (an implementation of ImageJ, a public domain program by W. Rasband available from http://rsb.info.nih.gov/ij/).

**Image analysis.** The analysis of smFISH images consisted of two components—segmentation and mRNA counting[13]. These two components were combined into an overall workflow that resulted in an image where each cell was annotated with the number of probes located in it. The image analysis workflow operated on image collections where each image represented a unique channel/z-stack pair. To separate the captured microscopy image into individual channel/z-stack pairs Bioformats was used, and the pipeline was implemented in the Python programming language.

**Quantification of 3′/5′ ratio for *COOLAIR* largest intron.** Images were acquired with the same laser intensity; exposure time and gain for the two probe sets (3′ probes were labelled with Quasar 670 and 5′ probes with Quasar 570—see Image Acquisition for specific lasers and filter sets used). Intensity of intronic signals was calculated from maximum intensity projection images, as total integrated intensity of each dot after background subtraction.

**Data availability.** The data that support the findings of this study are available from the corresponding author upon request.

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

## Acknowledgements

We thank Zhe Wu and Silvia Costa for comments on the manuscript, all members of the Dean and Howard groups for discussions and Mathew Hartley and Tjelvar Olsson for the help with the image analysis. This work was supported by the UK Biotechnology and Biological Sciences Research Council (BBSRC) grant BB/K00008X/1 and the Earth and Life Systems Alliance (a collaborative venture between John Innes Centre and University of East Anglia). C.D. acknowledges support from European Research Council Advanced grant MEXTIM and BBSRC Institute Strategic Programme grant BB/J004588/1.

## Author contributions

S.R., S.D. and C.D. conceived the study and designed the experiments; S.R. and S.D. performed the experiments; S.R. and C.D. and wrote the manuscript.

## Additional information

**Competing financial interests:** The authors declare no competing financial interests.

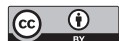

