## [Peer Review File · Nature Communications]

Reviewers' comments:

Reviewer #1 (Remarks to the Author):

It has previously been proposed that COOLAIR antisense transcripts play a role in the initial down regulation of FLC in response to prolonged exposure to low temperatures. However, the importance of COOLAIR in this process has been called into question because FLC repression is still observed on transgenes that lack a COOLAIR promoter, and at loci where COOLAIR transcription is disrupted by T-DNA insertion. In this elegant analysis using a combination of RNA and DNA FISH to determine the location of COOLAIR, and FLC transcripts as well as the FLC locus, Rosa et al clearly demonstrate a role for COOLAIR in facilitating the repression of FLC in the stele cells of Arabidopsis roots.

I enjoyed reading this manuscript, which presents some very elegant data that clearly warrants publication. The conclusions are generally well supported by the data presented and show a novel role of long non-coding RNAs in regulating gene expression in plants. The authors correctly point out that this situation is similar to the role of XIST in regulating X chromosome inactivation. I have only a few criticisms of the work as presented; these are outlined below for the authors' consideration.

1. Page 1 line 27; I suggest rewording this sentence as follows "with general implications for some other loci showing both sense and antisense transcription". I believe that the original version rather overstates the case as the authors themselves identify 3 loci showing sense and antisense transcription, but for only 1, the XIST transcript that plays a role in X inactivation, does the situation resemble that at FLC.
2. Page 1 line 37; " epigenetic silencing of FLC quantitatively accumulates" - I suggested using the word increases rather than accumulates simply because I find the notion of silencing "accumulating" a bit odd.
3. Please change the terminology for the nascent transcripts (ie that have not yet been processed) from unspliced to non-spliced throughout. The term unspliced implies that the transcripts were spliced that then this processing was reversed.
4. I found the concept of budding of COOLAIR transcripts described in Figure Extended data Figure 3b and the text on page 3 less than convincing.
5. I am also a bit confused about the role of spliced and nascent (non-spliced) COOLAIR transcripts. While I appreciate that it is not a trivial matter to distinguish between these I wonder if it would be possible to elucidate this more clearly. Are both the spliced and non-spliced forms present at the FLC locus? Are both present in the smaller foci of COOLAIR? What role do you think splicing of this transcript plays in its function?
6. I have one quibble with the use of root stele cells to develop the model that the rapid repression of FLC by COOLAIR allows induction of FT. While FT, the main target of FLC, is expressed predominantly in the vasculature of aerial tissue, it is not expressed in the vasculature of roots (Takada et al., 2003). Is there any evidence showing that the model presented here can be extrapolated to the vasculature of aerial tissue?
7. Extended data Fig 2; the legend indicates that signals from non-spliced FLC message appear in red - however there are no red signals in this image. Were the cells probed with the appropriate oligos to give a signal from non-spliced transcripts or is this an error in the legend?

8. There is no scale bar in Extended data Figure 5, contrary to the information in the legend.
9. Extended data Fig 7(c). Quantification of average mRNAs per cell in *ndx1-1* FRI mutant plants depicted in d. There is no d, so I suspect that this should be (b).
10. There seems to be a duplicated step in the M&M section describing combined RNA-DNA FISH P10 lines 371-374.

Reviewer #2 (Remarks to the Author):

The manuscript by Rosa et al. describes the mechanism by which the environmentally regulated long noncoding RNA COOLAIR mediates epigenetic repression of FLOWERING LOCUS C (FLC), which regulates flowering time in *Arabidopsis*. COOLAIR transcription is induced by cold treatment and leads to the accelerated repression of FLC through a switch in the histone modification state and recruitment of the Polycomb complex to maintain the silenced state. Here, the authors use single molecule in situ hybridization to demonstrate that COOLAIR transcripts, strongly upregulated by cold, form dense clouds covering the FLC loci (the authors make a parallel with the lncRNA Xist) and progressively lead to FLC silencing. Interestingly, the authors also show that, mechanistically, FLC sense and antisense transcription are mutually exclusive at individual loci, although these transcripts can accumulate in the same cell.

This is a short but elegant study that takes advantage of single molecule in situ hybridization analysis to examine the mechanistic relationship of FLC and COOLAIR transcription at the cellular level. I would support this manuscript for publication at Nature Communication after the authors have addressed the specific points listed below.

Specific comments:

1. The authors state that "The different dynamics of the unspliced and spliced RNA decay reveals that FLC transcription shutdown occurs rapidly after cold exposure, while the mRNA remains high for longer due to its relatively long half-life". I do not think the authors can conclude that the unspliced and spliced RNA have different decay dynamics because there are just too few unspliced FLC molecules at the non-vernalized stage (red foci detected in the NV panel Fig 2b). The dynamics of spliced and unspliced RNA foci disappearance might be the same when normalized to the total number of the red (unspliced) and green (spliced) foci (indeed there are only 2 unspliced FLC foci apparent in the NV image thus it is difficult to measure the decay dynamic). The authors should either show the normalized data for several cell images or remove their statement about the differences in decay dynamics and fast transcriptional shutdown at the FLC locus after cold exposure.
2. As the entire study is based on the single molecule in situ hybridization analysis, it is very important to distinguish between real and off target effects of single-molecule RNA FISH probes. The authors should include a control experiment with a different set of probes similar to what was done in Cabili et al., 2015, Genome Biol. This is particularly important for COOLAIR as it has been shown that "lncRNA FISH is prone to artifacts" due their repetitive sequence nature (Cabili et al., 2015). It should be possible to use two subsets,

'even' and 'odd', of probes, each labeled with a differently colored fluorophore. The same should also be done for the unspliced FLC transcript because of its low abundance and thus high probability to have off target/background dots.

Reviewer #3 (Remarks to the Author):

The antisense transcription of COOLAIR and COOLAIR transcripts play roles in epigenetic silencing of FLC in Arabidopsis in response to cold exposure. The authors applied single molecular FISH (smFISH) to dissect the mechanistic relationship of FLC and COOLAIR transcription at the single cell level. They found that the sense and antisense transcripts could co-occur in one cell but they were mutually exclusive at individual loci. Cold up-regulates COOLAIR transcription and through the mutually exclusive relationship that enhances shutdown of sense FLC transcription. Furthermore, they found that the accumulation of COOLAIR transcripts also suppressed FLC transcription after the cold exposure. This manuscript thus confirmed the view that the transcription of COOLAIR and COOLAIR transcripts both regulate FLC epigenetic silencing during vernalization using smFISH. Their data are clear and conclusions are robust. There are a few points the authors should address.

1. In lines 83 and 84, the authors hypothesized smaller foci of COOLAIR might correspond to intron lariats prior to degradation. This point needs to be further clarified because it is important to conclude the in cis accumulation of COOLAIR. Is there any experimental evidence to support this view? For instance, a dual-color smFISH with probes that recognize introns or exons of COOLAIR may reveal some clue? or knocking down the enzymes that are responsible for intron lariats debranching followed by smFISH may also confirm that the intron lariats are key components of these small foci?
2. In Extended Data Fig. 8. Is it sensitive enough to discriminate the two alternative scenarios for COOLAIR transcription (Bursty Transcription and Accumulation at this locus)? What is the transcription elongation speed at the FLC locus? What is the half-life of the unspliced COOLAIR and spliced COOLAIR?
3. In this manuscript the authors show a mutually exclusive sense-antisense transcription of COOLAIR and FLC as well as the importance of COOLAIR in the suppression of FLC transcription. More broadly, In order to enhance the significance of this finding I would posit that the bigger question is how many this type of regulation might be present in the Arabidopsis genome?

Reviewer #4 (Remarks to the Author):

Dean and colleagues report single molecule FISH studies of FLC sense and antisense transcription. The experimental design is elegant and the results are convincing. The in situ

and single molecule view of FLC mRNA and COOLAIR lncRNA, both arising from the FLC locus, during cold exposure has clarified and extended the mechanism of COOLAIR function. Specifically, this analysis has shown the mutually exclusive nature of the sense-antisense transcription; the switch is thrown per locus in the same nucleus. These results would be difficult to obtain without this in situ imaging approach. I support publication in Nature Communications. The following suggestions are meant to improve the clarity of the manuscript.

Minor comment:

1. Page 3 line 87. This statement is unclear. If NDX1 upregulates COOLAIR, why is COOLAIR level higher in ndx1-1 mutant? Is ndx1-1 a gain of function mutant? OR does R-loop repress COOLAIR transcription? This sentence should be rephrased to clarify meaning.

Reviewer #5 (Remarks to the Author):

The manuscript by Rosa et al. describes the role of antisense transcripts called COOLAIR in the cold-induced silencing of Arabidopsis of FLC gene. By adapting a single molecule fluorescence in situ hybridization (smFISH) technique in plants, the authors obtained an impressive signal to noise ratio to count the number of sense and antisense transcripts. First, they showed that FLC transcription is turned off within 2 weeks after cold exposure by imaging the unspliced nascent FLC mRNA. Then they used two-color smFISH to show that the sense (FLC) and antisense (COOLAIR) transcripts can occur in the same cell but they are mutually exclusive at individual loci. Finally, they confirmed that transcription of COOLAIR indeed had implications in FLC suppression by using FLC-TEX mutant line that is defective in cold induction of COOLAIR.

I think this is an elegant study that shows how single-cell single-molecule imaging technique can elucidate detailed mechanisms of gene silencing. Using conventional whole cell population data, it would be impossible to reveal the mutually exclusive transcription of the sense or antisense RNA at individual loci. On a technical side, their smFISH images are of high quality and very convincing. The authors used the technique effectively to answer their biological questions.

There is a lot of interest in the role of antisense transcripts in many different organisms. I find that it is interesting that COOLAIR in plants shows many parallels with the Xist RNA in mammals. While other antisense RNAs in yeast have different properties and functions, the antagonistic sense-antisense transcription seems to be evolutionally conserved in epigenetic mechanisms. I think this work will be of interest to the broad readership of Nature Communications. I have only some minor concerns, questions and suggestions as described below.

1. It would be helpful if the authors explain what R-loop is in this manuscript. It is difficult to understand the suggested molecular mechanism (lines 85-87 and extended data fig 7) without reading their previous paper on the R-loop. Perhaps in the first or second paragraph of the manuscript, the authors could summarize previous findings on the R-loop.

2. In line 66 on page 2, the authors say "The distribution is reminiscent of Cajal bodies..." Since there are various kinds of nuclear bodies, I suggest they change 'Cajal bodies' to 'nuclear bodies' in this sentence.

3. In line 79 on page 3, the authors described "The smaller COOLAIR foci appeared slightly later than the larger ones..." Because smFISH in fixed cells cannot report the time sequence of events, this sentence needs to be revised.

4. In line 124, I wonder if authors can provide the timescale of bulk intragenic Polycomb nucleation instead of just saying "much faster timescales". To me, 2 weeks sounds like a long time.

5. In lines 125-127, it is not clear why they think the two scenarios (transcriptional interference and double-stranded RNA mechanism) are unlikely. More explanation is needed to follow their reasoning.

Point by point discussion of the referees' comments

Reviewer#1

[COMMENT #1] Page 1 line 27; I suggest rewording this sentence as follows "with general implications for some other loci showing both sense and antisense transcription". I believe that the original version rather overstates the case as the authors themselves identify 3 loci showing sense and antisense transcription, but for only 1, the XIST transcript that plays a role in X inactivation, does the situation resemble that at FLC.

[RESPONSE #1] we have changed the text as suggested.

"These results have general implications for other loci showing both sense and antisense transcription."

[COMMENT #2] Page 1 line 37; " epigenetic silencing of FLC quantitatively accumulates" - I suggested using the word increases rather than accumulates simply because I find the notion of silencing "accumulating" a bit odd.

[RESPONSE #2] we have changed the text as suggested.

[COMMENT #3] Please change the terminology for the nascent transcripts (ie that have not yet been processed) from unspliced to non-spliced throughout. The term unspliced implies that the transcripts were spliced that then this processing was reversed.

[RESPONSE #3] unspliced has been replaced with non-spliced throughout.

[COMMENT #4] I found the concept of budding of COOLAIR transcripts described in Figure Extended data Figure 3b and the text on page 3 less than convincing.

[RESPONSE #4] we have removed the comment- possibly splitting from the larger foci.

[COMMENT #5] I am also a bit confused about the role of spliced and nascent (non-spliced) COOLAIR transcripts. While I appreciate that it is not a trivial matter to distinguish between these I wonder if it would be possible to elucidate this more clearly. Are both the spliced and non-spliced forms present at the FLC locus? Are both present in the smaller foci of COOLAIR? What role do you think splicing of this transcript plays in its function?

[RESPONSE #5] Probes targeting *COOLAIR* exonic sequences were only detected at the locus, most likely corresponding to the nascent transcript. We have added *"Probes to COOLAIR exonic sequences did not reveal signals in the cytoplasm (Supplementary Fig. 6b). We have previously found spliced COOLAIR enriched in cytoplasmic fractions⁷, so the absence of cytoplasmic signal in the FISH may be due to extensive secondary structure of the spliced COOLAIR transcripts^{15"}*.

We have also used dual-color smFISH probes to detect introns or exons of *COOLAIR* (Supplementary Fig.6a,c). The smaller foci corresponded mostly to intronic signal.

[COMMENT #6] I have one quibble with the use of root stele cells to develop the model that the rapid repression of FLC by COOLAIR allows induction of FT. While FT, the main target of FLC, is expressed predominantly in the vasculature of aerial tissue, it is not expressed in the

vasculature of roots (Takada et al., 2003). Is there any evidence showing that the model presented here can be extrapolated to the vasculature of aerial tissue?

[RESPONSE #6] We have removed the statement "Finally, *COOLAIR* upregulation in the cold occurs mostly in provasculature cells. Given that *FLOWERING LOCUS T (FT)*, one of the main targets of *FLC*, is expressed in the vasculature²⁰ and since *FLC* does not move between cells⁹, this pattern of expression for *COOLAIR* might be especially important for the control of *FLC* expression in this tissue." We have been unable to visualise a FISH signal in aerial parts of the plant due to the very high background from chlorophyll fluorescence, so we agree that without that data it is better not to comment on FT.

[COMMENT #7] Extended data Fig 2; the legend indicates that signals from non-spliced FLC message appear in red - however there are no red signals in this image. Were the cells probed with the appropriate oligos to give a signal from non-spliced transcripts or is this an error in the legend?

[RESPONSE #7] This was an error in the figure legend. We have now corrected this error.

[COMMENT #8] There is no scale bar in Extended data Figure 5, contrary to the information in the legend.

[RESPONSE #8] We have now added the scale bars.

[COMMENT #9] Extended data Fig 7(c). Quantification of average mRNAs per cell in *ndx1-1 FRI* mutant plants depicted in d. There is no d, so I suspect that this should be (b).

[RESPONSE #9] We have corrected this error.

[COMMENT #10] There seems to be a duplicated step in the M&M section describing combined RNA-DNA FISH P10 lines 371-374.

[RESPONSE #10] We have corrected this error.

Reviewer#2

Major points:

[COMMENT #1] The authors state that "The different dynamics of the unspliced and spliced RNA decay reveals that FLC transcription shutdown occurs rapidly after cold exposure, while the mRNA remains high for longer due to its relatively long half-life". I do not think the authors can conclude that the unspliced and spliced RNA have different decay dynamics because there are just too few unspliced FLC molecules at the non-vernalized stage (red foci detected in the NV panel Fig 2b). The dynamics of spliced and unspliced RNA foci disappearance might be the same when normalized to the total number of the red (unspliced) and green (spliced) foci (indeed there are only 2 unspliced FLC foci apparent in the NV image thus it is difficult to measure the decay dynamic). The authors should either show the normalized data for several cell images or remove their statement about the differences in decay dynamics and fast transcriptional shutdown at the FLC locus after cold exposure.

[RESPONSE #1] We have removed this statement.

[COMMENT #2] As the entire study is based on the single molecule in situ hybridization analysis, it is very important to distinguish between real and off target effects of single-molecule RNA FISH probes. The authors should include a control experiment with a different set of probes similar to what was done in Cabili et al., 2015, Genome Biol. This is particularly important for COOLAIR as it has been shown that "lncRNA FISH is prone to artifacts" due their repetitive sequence nature (Cabili et al., 2015). It should be possible to use two subsets, 'even' and 'odd', of probes, each labeled with a differently colored fluorophore. The same should also be done for the unspliced FLC transcript because of its low abundance and thus high probability to have off target/background dots.

[RESPONSE #2] We agree with the reviewer that it is important to confirm the specificity of the probes used. While we did not use even-odd probe sets to check for specificity we have used two different subsets of probes for COOLAIR and in each case found co-localization - the 5'/3' set in the large intron (Fig.2 f-h) and the exonic+intronic set (Supplementary Fig. 6a,c). The regulation of the expression in different genotypes and environments also shows the probes are specific: 1) almost no signal in NV, up-regulation in the cold and then off again at later time points; 2) very low signal in TEX577; 3) up-regulation in *ndx* mutant. Importantly we also observed co-localization of the FLC DNA FISH with COOLAIR RNA FISH signals (Fig. 2d).

For the case of FLC unspliced transcripts we have also a consecutive 5' and 3' probe sets for the large intron one showing co-localization – we have included this additional data in Supplementary Fig. 2c. Additionally, we have observed that FLC unspliced transcripts co-localize with FLC DNA FISH signals (Supplementary Fig. 2d), which again confirms the specificity of these probes. We have also emphasised these points in the main text.

Reviewer#3

[COMMENT #1] In lines 83 and 84, the authors hypothesized smaller foci of COOLAIR might correspond to intron lariats prior to degradation. This point needs to be further clarified because it is important to conclude the in cis accumulation of COOLAIR. Is there any experimental evidence to support this view? For instance, a dual-color smFISH with probes that recognize introns or exons of COOLAIR may reveal some clue? or knocking down the enzymes that are responsible for intron lariats debranching followed by smFISH may also confirm that the intron lariats are key components of these small foci?

[RESPONSE #1] We have used dual-color smFISH probes to detect introns or exons of COOLAIR (Supplementary Fig.6). Indeed the smaller foci corresponded **mostly** to intronic signal.

[COMMENT #2] In Extended Data Fig. 8. Is it sensitive enough to discriminate the two alternative scenarios for COOLAIR transcription (Bursty Transcription and Accumulation at this locus)? What is the transcription elongation speed at the FLC locus? What is the half-life of the unspliced COOLAIR and spliced COOLAIR?

[RESPONSE #2] We had previously measured the elongation rate for FLC/COOLAIR transcription in a FRI+ genotype as on the order of 2-3kb/min, a fairly average rate (Wu et al PNAS 2016). We

are in the process of measuring *COOLAIR* half-life. In order to address the reviewer's concern we have edited this section extensively.

[COMMENT #3] In this manuscript the authors show a mutually exclusive sense-antisense transcription of *COOLAIR* and *FLC* as well as the importance of *COOLAIR* in the suppression of *FLC* transcription. More broadly, In order to enhance the significance of this finding I would posit that the bigger question is how many this type of regulation might be present in the *Arabidopsis* genome?

[RESPONSE #3] Indeed, this is a really interesting question and awaits a detailed analysis of many more genes showing sense-antisense transcription using smFISH.

Reviewer#4

[COMMENT #1] Page 3 line 87. This statement is unclear. If *NDX1* upregulates *COOLAIR*, why is *COOLAIR* level higher in *ndx1-1* mutant? Is *ndx1-1* a gain of function mutant? OR does R-loop repress *COOLAIR* transcription? This sentence should be rephrased to clarify meaning.

[RESPONSE #1] We have rephrased this section more accurately.

“COOLAIR levels were higher in ndx1-1, a mutant defective in the homeodomain protein that represses COOLAIR transcription through stabilization of the R-loop at the COOLAIR promoter⁶ supporting the view that cold and NDX1 regulate COOLAIR transcription independently (Supplementary Fig. 7).”

Reviewer#5

[COMMENT #1] It would be helpful if the authors explain what R-loop is in this manuscript. It is difficult to understand the suggested molecular mechanism (lines 85-87 and extended data fig 7) without reading their previous paper on the R-loop. Perhaps in the first or second paragraph of the manuscript, the authors could summarize previous findings on the R-loop.

[RESPONSE #1] We have added a sentence to this effect in the first paragraph of the main text.

“Prior to cold COOLAIR transcription is repressed by the NDX1 homeodomain protein, which stabilizes an R-loop (where the nascent transcript invades the DNA helix) over the COOLAIR promoter⁶.”

[COMMENT #2] In line 66 on page 2, the authors say "The distribution is reminiscent of Cajal bodies..." Since there are various kinds of nuclear bodies, I suggest they change 'Cajal bodies' to 'nuclear bodies' in this sentence.

[RESPONSE #2] We have replaced the text as suggested.

[COMMENT #3] In line 79 on page 3, the authors described "The smaller COOLAIR foci appeared slightly later than the larger ones..." Because smFISH in fixed cells cannot report the time sequence of events, this sentence needs to be revised.

[RESPONSE #3] We have removed this sentence.

[COMMENT #4] In line 124, I wonder if authors can provide the timescale of bulk intragenic Polycomb nucleation instead of just saying "much faster timescales". To me, 2 weeks sounds like a long time.

[RESPONSE #4] We have added specific timescales for transcriptional shut-down as compared to nucleation.

"COOLAIR up-regulation coincides with the disruption of a gene loop at FLC¹⁹ and accelerates sense transcriptional shutdown within 1-2 weeks as compared to the 4 weeks required for maximal intragenic Polycomb nucleation⁸."

[COMMENT #5] In lines 125-127, it is not clear why they think the two scenarios (transcriptional interference and double-stranded RNA mechanism) are unlikely. More explanation is needed to follow their reasoning.

[RESPONSE #5] We have edited this section and provided an additional reference to clarify what we mean.

"The anti-correlation of COOLAIR and FLC transcription at the gene level would seem to exclude a cis-based colliding polymerase transcriptional interference mechanism. There is also no evidence for double-stranded RNA and the RNAi machinery being required for vernalization²⁰."

REVIEWERS' COMMENTS:

Reviewer #1 (Remarks to the Author):

The authors have addressed most of my criticisms during the revision of this manuscript but I regret to say that I still have some criticisms, which I have outlined below for the authors' consideration.

1. Page 1 line 19; Although the authors have changed the wording for this sentence, on re-reading this I still think overstates the case. I suggest "These results may have general implications for other loci showing both sense and antisense transcription" would be better.
2. Supplementary Figure 6; The images in panels b and c have the label "COOLAIR Spliced". As far as I can tell from the location of probes in panel a, it is not possible to determine if the transcript has been spliced just by probing with the green (exonic) probes. This is confirmed by the colocalization of exonic and intronic probes for some of the signals in panel c indicating that non-spliced transcripts also give a signal with the exonic probes as one would expect. This means that the panels labelled "Spliced" show a mix of spliced and non-spliced transcripts.
3. Supplementary Figure 6; It also seems that there is an error in the legend for panel c as non-spliced is listed as a green signal in the legend although for both the legend for panel a and the Figure itself, non-spliced is listed as a red signal. Which is correct, the figure or the legend for c? The term "a different pattern of co-localization was observed for smaller foci" is rather meaningless as there is no co-localization in the smaller foci, or at least not those depicted in this figure.
4. Supplementary Figure 7; The legend for panel a indicates the time NV, 1WV, 2WV and 3WV, but there is no sample for 3WV.
5. Supplementary Figure 7; I don't understand the significance of panel b, which is listed as showing the frequency distribution of FLC mRNA molecules per cell in non-vernalized *ndx1-1* FRI mutant plants. Is this to be compared with the data presented in Figure 1c? If so it would be useful to have the average number of mRNA molecules provided for the data in 1(c).
6. There is an inconsistency in the citations for the additional data; most of these figures are now referred to as Supplementary Figure x, with the exception of Extended Data 9 (p4 line 119).

Reviewer #2 (Remarks to the Author):

All of my concerns have been addressed. I would recommend this manuscript for publication.

Reviewer #3 (Remarks to the Author):

My comments have been addressed. No further revisions required.

Reviewer #5 (Remarks to the Author):

The revised manuscript addressed all my previous concerns and questions.

Responses to reviewer#1:

Comment 1. Page 1 line 19; Although the authors have changed the wording for this sentence, on re-reading this I still think overstates the case. I suggest “These results may have general implications for other loci showing both sense and antisense transcription” would be better.

Response 1. We have now replaced the statement as suggested by the reviewer.

Comment 2. Supplementary Figure 6; The images in panels b and c have the label “COOLAIR Spliced”. As far as I can tell from the location of probes in panel a, it is not possible to determine if the transcript has been spliced just by probing with the green (exonic) probes. This is confirmed by the colocalization of exonic and intronic probes for some of the signals in panel c indicating that non-spliced transcripts also give a signal with the exonic probes as one would expect. This means that the panels labelled “Spliced” show a mix of spliced and non-spliced transcripts.

Response 2. The reviewer is absolutely correct. The use of the word “spliced” is inaccurate since these probes will also hybridize with the nascent transcript. We therefore replaced “spliced” by **exonic** and “non-spliced” by **intronic** when mentioning smFISH probes.

Comment 3. Supplementary Figure 6; It also seems that there is an error in the legend for panel c as non-spliced is listed as a green signal in the legend although for both the legend for panel a and the Figure itself, non-spliced is listed as a red signal. Which is correct, the figure or the legend for c? The term “a different pattern of co-localization was observed for smaller foci” is rather meaningless as there is no co-localization in the smaller foci, or at least not those depicted in this figure.

Response 3. Indeed, there was a mistake in the legend of Supplementary Figure 6 and we are grateful to the reviewer’s diligence for spotting this. In panel c, COOLAIR “non-spliced (now intronic)” is in red and COOLAIR “spliced (now exonic)” is in green. We have now corrected this mistake.

Comment 4. Supplementary Figure 7; The legend for panel a indicates the time NV, 1WV, 2WV and 3WV, but there is no sample for 3WV.

Response 4. We have now removed the reference to time point 3W, which does not exist in the graph.

Comment 5. Supplementary Figure 7; I don’t understand the significance of panel b, which is listed as showing the frequency distribution of FLC mRNA molecules per cell in non-vernalized *ndx1-1* FRI mutant plants. Is this to be compared with the data presented in Figure 1c? If so it would be useful to have the average number of mRNA molecules provided for the data in 1(c).

Response 5. The data from *ndx* mutant was included to support specificity of the probes used for *COOLAIR* – as *COOLAIR* is up-regulated in this mutant. We included the data from FLC sense mRNAs per cell (NV) for completeness but we did not intend to make any further conclusions from these data. We have now simply removed panel b and c.

Comment 6. There is an inconsistency in the citations for the additional data; most of these figures are now referred to as Supplementary Figure x, with the exception of Extended Data 9 (p4 line 119).

Response 6. We have now replaced “Extended Figure 9” by Supplementary Figure 9.